# Collaborative Generative AI: Integrating GPT-$k$ for Efficient Editing in Text-to-Image Generation

**Wanrong Zhu[¶], Xinyi Wang[¶], Yujie Lu[¶], Tsu-Jui Fu[¶],**
**Xin Eric Wang[§], Miguel Eckstein[¶], William Yang Wang[¶]**
[¶]UC Santa Barbara, [§]UC Santa Cruz
{wanrongzhu, xinyi_wang, yujielu, tsu-juifu, migueleckstein, william}@ucsb.edu, xwang366@ucsc.edu

## Abstract

The field of text-to-image (T2I) generation has garnered significant attention both within the research community and among everyday users. Despite the advancements of T2I models, a common issue encountered by users is the need for repetitive editing of input prompts in order to receive a satisfactory image, which is time-consuming and labor-intensive. Given the demonstrated text generation power of large-scale language models, such as GPT-$k$, we investigate the potential of utilizing such models to improve the prompt editing process for T2I generation. We conduct a series of experiments to compare the common edits made by humans and GPT-$k$, evaluate the performance of GPT-$k$ in prompting T2I, and examine factors that may influence this process. We found that GPT-$k$ models focus more on inserting modifiers while humans tend to replace words and phrases, which includes changes to the subject matter. Experimental results show that GPT-$k$ are more effective in adjusting modifiers rather than predicting spontaneous changes in the primary subject matters. Adopting the edit suggested by GPT-$k$ models may reduce the percentage of remaining edits by 20-30%.

## 1 Introduction

The task of text-to-image (T2I) generation, which involves generating images from natural language descriptions, holds significant potential to create new avenues and job opportunities for content creators while also providing insights into the grounding of natural language in the visual world through the application of generative AI. A number of models have demonstrated exceptional performance in terms of image quality, such as StableDiffusion (Rombach et al., 2021), Midjourney (Midjourney, 2022), Imagen (Saharia et al., 2022), and DALLE-2 (Ramesh et al., 2022). Despite the popularity of T2I generation and the ability of these models to generate impressive images, the difficulty of

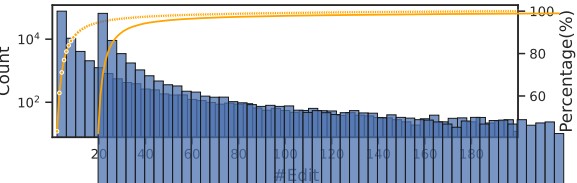

Figure 1: The histogram plots #edit per trace in DiffusionDB (Wang et al., 2022) while the lineplot shows the cumulative percentage of traces less than certain #edit. The y-axis on the left is on log-scale. On average, there are 6.9 edits being made in each user editing trace.

"prompt-engineering" – which is writing accurate prompts to describe the desired image in this scenario – still remains a significant challenge. Users often need to edit the prompt for several rounds before arriving at a satisfactory image, which makes the process of T2I generation time-consuming and laborious (and expensive for commercialized models). Figure 1 shows the #edits distribution in the editing traces made by StableDiffusion Discord server users (Wang et al., 2022). This phenomenon highlights the need to improve the efficiency and effectiveness of prompting T2I models.

Recently, large language models (LLMs) such as GPT-$k$ (Radford et al., 2019; Brown et al., 2020; Ouyang et al., 2022) have demonstrated impressive abilities in text generation. This leads to the natural question of whether these models can be utilized to improve the T2I prompting process (Chakrabarty et al., 2023). However, LLMs and T2I models have different architectures and are often trained on different modalities, which makes it challenging to integrate them seamlessly.

In this paper, we conduct a series of experiments and analyses on user editing traces on StableDiffusion,[1] and attempt to modify the T2I prompts with eight GPT-$k$ models. The primary objective of our research is to examine the potential of modifying T2I prompts with GPT-$k$ models. More specifically,

---

[1]Our experiments are conducted upon StableDiffusion since it is a wide-adopted open-source large text-to-image generative model with SoTA performance.

we aim to investigate the common edits made by humans and by GPT-$k$, as well as the performance of prompting T2I generation with GPT-$k$ models. Additionally, we aim to identify and investigate possible factors that might influence the performance of GPT-$k$ in T2I generation tasks.

Through our experiments, we observe that:

- While GPT-$k$ models tend to focus more on inserting modifiers, human users have a greater tendency towards replacing words and phrases, which may include spontaneous changes to the subject matter of the prompt.
- Modifying the T2I prompt with GPT-$k$ models may not necessarily result in a direct increase in similarity to the final target image in the editing trace. Instead, the edit suggested by GPT-$k$ may be closely related to intermediate editing steps, and the percentage of remaining edits may decrease by 20-30% if the edit suggested by GPT-$k$ is adopted.
- These findings suggest that instead of attempting to predict spontaneous changes made by human users on the primary subject matter, GPT-$k$ models are more effective in improving prompts by rewriting and performing edits related to modifiers adjustment.

## 2 Research Questions and Settings

To investigate the potential of modifying T2I prompts with GPT-$k$ models, we conduct a series of experiments and analysis, aiming to answer the following questions:

1. *To what extent can GPT-$k$ models help prompt text-to-image generation?*
2. *What are the common types of edits made by humans and by GPT-$k$ models?*
3. *What are the factors that may influence GPT-$k$ prompting text-to-image generation?*

We describe the dataset, models and evaluation metrics used in this study below.

**Dataset** DiffusionDB-2M (Wang et al., 2022) scrapes 2M groups of user prompts, hyperparameters and images generated by StableDiffusion (Rombach et al., 2021) from the official Stable Diffusion Discord server. We group the prompts by anonymized user_id, then cluster the user prompts into *traces of edits* based on the prompt contents. More specifically, we encode prompts with Universal Sentence Encoder (Cer et al., 2018), then cluster upon the prompt embeddings with DB-SCAN (Ester et al., 1996). The order of edits within

| GPT-$k$ | Model Name | #Parameters |
|---|---|---|
| GPT-2 (Radford et al., 2019) | gpt2-base | 117M |
| | gpt2-medium | 345M |
| | gpt2-large | 774M |
| | gpt2-xl | 1.2B |
| GPT-3 (Brown et al., 2020) | text-ada-001 | 350M |
| | text-babbage-001 | 1.3B |
| | text-curie-001 | 6.7B |
| GPT-3.5 (Ouyang et al., 2022) | text-davinci-003 | 175B |

Table 1: The names and corresponding parameter sizes of the GPT-$k$ models covered in our study.

each trace is determined by the timestamps. We receive 100k traces of edits, the mean #edit per trace is 6.9, with a standard deviation of 16.1. Figure 1 plots the #edit in the clustered traces, which shows a long-tail distribution with about 5% of the traces having more than 20 edits. Thus, in the following experiments, the evaluation results were reported on traces with at most 20 edits.

**GPT-$k$ & Text-to-Image Models** Table 1 lists the names and parameter sizes of the eight GPT-$k$ models we covered in this study. We conduct T2I experiments with the open-source StableDiffusion-v1-4 (Rombach et al., 2021), which is used to render images in DiffusionDB (Wang et al., 2022).

**Annotations** For each trace of length $n$, we denote the prompts as $(t_1, t_2, \ldots, t_n)$, and the generated images as $(i_1, i_2, \ldots, i_n)$. We refer to the GPT-$k$-modified prompt as $t'$, and refer to the image rendered from the modified prompt as $i'$.

**Metrics** We use the cosine similarity of CLIP-ViT/B-32 (Radford et al., 2021) visual features to evaluate the similarity between images. Let $i_k$ denote the image in the original edit trace that is most similar to $i'$, we define the Relative Number of Edits (RNE) as $\frac{n-k}{n-1}$. The RNE metric reflects the relative position of the edit in the original trace that is most similar to the edit suggested by GPT-$k$ and also represents the percentage of remaining edits after the edit suggested by GPT-$k$ is performed.

## 3 Prompting T2I w/ GPT-$k$

In the following experiments, we only reveal the initial prompts in the user editing traces to GPT-$k$, and ask the models to perform *one* round of edit.

**Procedure** We split the 100k trace of edits into two parts, with 30k traces used for evaluations and the remaining 70k serving as heldout set. For each of the listed GPT-$k$ models, we provide the first prompt $t_1$ in each evaluation trace to the model, and let GPT-$k$ generate the modified prompt $t'$.

GPT-2 models are finetuned with the prompts in

| Model | $\boldsymbol{i}_1$-$\boldsymbol{i}_n$ | $\boldsymbol{i}_{n-1}$-$\boldsymbol{i}_n$ | $\boldsymbol{i}'$-$\boldsymbol{i}_n$ | $\boldsymbol{i}'$-$\boldsymbol{i}_{MS}$ | RNE(%) |
|---|---|---|---|---|---|
| gpt2-base | | | 69.58 | 80.16 | 75.28 |
| gpt2-medium | | | 69.63 | 80.39 | 78.28 |
| gpt2-large | | | 69.70 | 80.50 | 78.88 |
| gpt2-xl | 71.72 | 74.82 | 69.57 | 80.37 | 75.25 |
| gpt3-ada | | | 66.95 | 76.37 | 69.43 |
| gpt3-babbage | | | 68.79 | 78.81 | 72.33 |
| gpt3-curie | | | 68.45 | 78.28 | 71.41 |
| gpt3.5-davinci | | | 68.79 | 78.09 | 69.22 |

Table 2: The CLIP cosine similarity scores between images and the relative number of edits (RNE). Here, $\boldsymbol{i}_1$, $\boldsymbol{i}_{n-1}$, $\boldsymbol{i}_n$ denotes the first, last-but-one, and last image in the trace of edits; $\boldsymbol{i}'$ is the image generated from the modified prompt, and $\boldsymbol{i}_{MS}$ is the image that is most similar to $\boldsymbol{i}'$ with regard to CLIP cosine similarity. RNE scores suggest a 20-30% decrease in the percentage of remaining edits if adopting edits suggested by GPT-$k$.

the heldout traces for two epochs with a learning rate of $5e-5$ and a batch size of 128. For GPT-3/3.5 models, an in-context learning approach is adopted. Following previous studies (Yang et al., 2021; Alayrac et al., 2022), supporting examples for in-context learning are selected by comparing the similarity of the prompt text features and retrieving $(\hat{\boldsymbol{t}}_1, \hat{\boldsymbol{t}}_n)$ pairs from the top-$m$ most similar traces. Modified prompts are generated through 16-shot in-context learning with $m$=16. Appendix Table 8 shows how we prompt GPT-3/3.5.

After receiving the GPT-$k$-modified prompt $\boldsymbol{t}'$, we provide it to StableDiffusion to generate image $\boldsymbol{i}'$. The effectiveness of GPT-$k$ in prompting T2I generation is evaluated by comparing the similarity of the generated image $\boldsymbol{i}'$ to images in the original trace. The results reported in the following sections are the mean of three repeated runs.

**Automatic Evaluation** The CLIP cosine similarity scores, as listed in Table 2, are used to evaluate image similarity. Two model-agnostic baselines are established for comparison: the similarity between the first and last images ($\boldsymbol{i}_1$-$\boldsymbol{i}_n$), and the similarity between the last-but-one and last images ($\boldsymbol{i}_{n-1}$-$\boldsymbol{i}_n$).

We denote $\boldsymbol{i}_{MS}$ as the image in the original trace that is most similar to the generated image $\boldsymbol{i}'$ (has the highest CLIP cosine similarity score). Examining results in Table 2, we notice that the image $\boldsymbol{i}'$ generated from the modified prompt may not be directly similar to the final target $\boldsymbol{i}_n$, as ($\boldsymbol{i}'$-$\boldsymbol{i}_n$) is lower than the baselines. However, it appears that $\boldsymbol{i}'$ may be related to the intermediate steps in the editing trace, as evidenced by the significantly higher similarity between $\boldsymbol{i}'$ and $\boldsymbol{i}_{MS}$ compared to the baselines. RNE scores show that, $\boldsymbol{i}'$ is most similar to images in the first one-third of the trace, and

| | Effectiveness | | | Likelihood | | |
|---|---|---|---|---|---|---|
| | Win(%) | Tie(%) | Lose(%) | Win(%) | Tie(%) | Lose(%) |
| gpt2-xl | 38.57 | 29.77 | 31.66 | 38.99 | 22.01 | 38.99 |
| gpt3-curie | 40.89 | 21.33 | 37.78 | 39.33 | 21.56 | 39.11 |
| gpt3.5-davinci | 39.58 | 21.67 | 38.75 | 38.13 | 25.00 | 36.87 |

Table 3: Human evaluation results of head-to-head comparison between edits suggested by GPT-$k$ and human-made edits. We evaluate the effectiveness of the edit and the likelihood of this edit being adopted by humans. "Win" and "Tie" indicate that GPT-$k$-suggested edits are better than or comparable to human edits.

| | | Insert | Delete | Swap | Replace |
|---|---|---|---|---|---|
| Human | - | | 29.9% | 21.1% | 2.0% | 47.0% |
| GPT-$k$ | gpt2-base | 95.4% | 0.0% | 0.0% | 4.6% |
| | gpt2-medium | 95.6% | 0.1% | 0.0% | 4.4% |
| | gpt2-large | 95.0% | 0.1% | 0.0% | 4.9% |
| | gpt2-xl | 95.9% | 0.0% | 0.0% | 4.1% |
| | gpt3-ada | 36.5% | 14.9% | 2.0% | 46.6% |
| | gpt3-babbage | 39.6% | 18.8% | 3.3% | 38.3% |
| | gpt3-curie | 42.9% | 17.7% | 4.1% | 35.3% |
| | gpt3.5-davinci | 68.5% | 3.9% | 2.3% | 25.3% |

Table 4: The distribution of the common types of edits made by human and by GPT-$k$ models.

that the percentage of remaining edits decreases by 20-30% if the edit suggested by GPT-$k$ is adopted.

**Human Evaluation** For each editing trace, we present MTurk annotators with the initial prompt and image ($\boldsymbol{t}_1$, $\boldsymbol{i}_1$), and two candidate edits: (1) the GPT-$k$-modified prompt, $\boldsymbol{t}'$; (2) the human edit, $\boldsymbol{t}_{MS}$, from the original editing trace. Here, $\boldsymbol{t}_{MS}$ is the corresponding prompt to $\boldsymbol{i}_{MS}$, which has the highest CLIP cosine similarity with $\boldsymbol{i}'$. The annotators are then asked to decide which edit was more effective and more likely to be adopted by human editors, $\boldsymbol{t}'$ or $\boldsymbol{t}_{MS}$. We evaluate each listed GPT-$k$ model with 200 traces. For each trace, three annotators are invited to provide their judgments.

As shown in Table 3, the three GPT-$k$ models all show tight wins against the human edits regarding both effectiveness and likelihood of being adopted. This verifies that the edits made by GPT-$k$ models are similar or comparable to the intermediate steps in the human editing trace.

## 4 Human's Common Edits vs. GPT-$k$'s

Upon examination of the user editing traces, we identify four types of edits commonly made by humans: (1) **Insert**: adding descriptive terms such as modifiers to the prompt to specify the style, artistic technique, camera view, lighting, etc; (2) **Delete**: removing certain terms; (3) **Swap**: changing the order of certain terms in the prompt; (4) **Replace**: changing the modifiers or the main subject of the

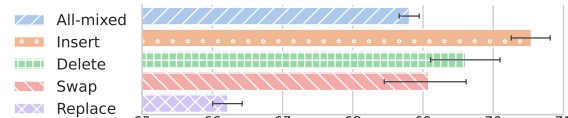

Figure 2: The CLIP cosine similarity scores between the image generated from the GPT-3.5-modified prompt and the last image. Results are reported on $\mathcal{S}_{\text{eval}}$ (All-mixed), $\mathcal{S}_{\text{insert}}$, $\mathcal{S}_{\text{delete}}$, $\mathcal{S}_{\text{swap}}$ and $\mathcal{S}_{\text{replace}}$.

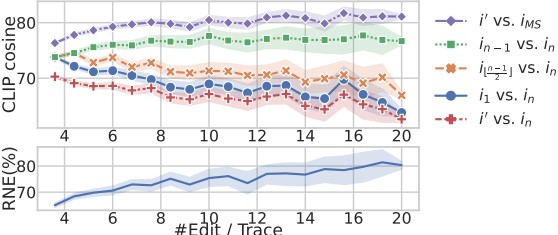

Figure 3: The CLIP cosine similarity (upper) and RNE (lower) on traces with #edit ranging from 2 to 20. Prompts for T2I generation are modified with GPT-3.5. Here, $i_1$, $i_{\lfloor \frac{n-1}{2} \rfloor}$, $i_{n-1}$, $i_n$ denotes the first, middle, last-but-one, and last image in the trace of edits; $i'$ is the image generated from the modified prompt, and $i_{MS}$ is the image that is most similar to $i'$.

prompt. Appendix Table 5 shows edit examples.

To count the frequency of edits, we first remove punctuation marks and stopwords using NLTK (Bird et al., 2009). We then utilize the SequenceMatcher[2] to compare the adjacent prompts in the trace and detect the edit type. Table 4 lists the frequency of common edits made by humans and by GPT-$k$ models. We notice a discrepancy between the distribution of human edits and the ones made by GPT-$k$. Nearly half of human edits pertain to replace, followed by insert and delete. GPT-2 models, due to their autoregressive training nature, have a tendency towards continual generation, resulting in a majority of edits being insert. While GPT-3/3.5 also undergoes autoregressive training, its extensive training data and enlarged model size empower it for emergent ability. Unlike GPT-2, which solely receives the specific prompt requiring editing during testing, GPT-3/3.5 is also provided with additional editing instances as supportive exemplars for in-context learning. Consequently, while GPT-2 indeed adheres to its autoregressive inference manner, the emergent capability of GPT-3/3.5 enables it to attempt other edit types that simulate human-like editing behavior. For GPT-3, as the model size increases, we see an increase in the frequency of insert and a decrease in replace. For GPT-3.5, the most frequent edit is insert, followed by replace; while delete and swap are relatively rare.

## 5 Ablations & Analyses

**Effects of Edit Types**  To investigate the effects of each individual edit type $e_i$, we create four subsets $\mathcal{S}_{\text{insert}}$, $\mathcal{S}_{\text{delete}}$, $\mathcal{S}_{\text{swap}}$ and $\mathcal{S}_{\text{replace}}$ from the evaluation set $\mathcal{S}_{\text{eval}}$. Each subset $\mathcal{S}_{e_i}$, comprises of traces that meet the criteria of "if and only if edit $e_i$ is applied on the first prompt can we receive the last prompt." For each edit type $e_i$, the image similarity between the image generated from the modified prompt and the last image for traces in $\mathcal{S}_{e_i}$ is calculated and

[2]https://docs.python.org/3/library/difflib.html

compared to the baseline results obtained from the complete $\mathcal{S}_{\text{eval}}$ that mixes all types of edits.

Figure 2 illustrates the impact of different edit types on image similarity. The CLIP cosine similarities of traces that solely consist of insert, delete, and swap edits are higher or comparable to the all-mixed baseline. This suggests that GPT-$k$ performs better at adding, removing, and reordering modifiers. Conversely, we observe that replace edits lead to lower image similarities. This is likely due to the fact that the replace edit sometimes results in a change of the subject matter, which can drastically alter the painting. It is worth noting that shifting the primary subject matter of the painting is a relatively spontaneous action made by humans. Appendix Table 6 shows a trace with multiple replace operation. The vast number of potential replacements makes it particularly difficult for GPT-$k$ to accurately select the desired edit.

**Effects of #Edits**  Figure 3 depicts how the CLIP cosine similarity changes with the #edit in the trace. As #edit increases, the similarity between the last image $i_n$ and those at the beginning or middle of the trace ($i_1$ or $i_{\lfloor \frac{n-1}{2} \rfloor}$) decreases. This trend may be attributed to the higher likelihood of changing primary subject matters in longer traces. Meanwhile, the similarity between the last-but-one and the last images ($i_{n-1}$ vs. $i_n$) remains consistent, suggesting that the majority of edits made towards the end of the prompt editing process are minor in nature. As the trace of edits gets longer, the similarity between the image $i'$ generated from the modified prompt and the most similar image $i_{MS}$ in the trace remains constant, while the RNE metric gradually increases to approximately 80%. This indicates that the modified prompt is related

to the early edits in the trace. This aligns with our previous findings, which suggest that GPT-$k$ is proficient in rewriting prompts and adjusting modifiers, but may struggle to predict spontaneous changes in the main subject matter of the painting.

## 6 Conclusion

Through our experiments and analyses, we hope to gain a deeper understanding of the capabilities and limitations of using GPT-$k$ to edit prompts for text-to-image generation, and provide valuable insights for future research in this field.

## Limitations

In this work, we are focusing on the StableDiffusion model for text-to-image generation. However, we have not examined user traces from other online text-to-image generation servers such as Midjourney or DALLE-2, as these models have not been publicly released and therefore cannot be replicated locally for experimentation.

Additionally, the current experimental setup only uses GPT-$k$ for a single round of editing, and future work should include human studies that use GPT-$k$ to suggest edits for multiple turns, and compare these to user traces without the use of GPT-$k$.

Furthermore, the current implementation only utilizes the initial prompt as input to GPT-$k$. In future studies, it would be beneficial to provide additional information to the GPT-$k$ models, including additional steps of editing or the generated image, in order to receive more specific feedback on the editing suggestions.

## Ethics Statement

The field of text-to-image generation is of great interest to the broad public, and every content creator should have the opportunity to explore its potential. However, the current prompt engineering process can be time-consuming and expensive for content creators, and the repetitive editing and regeneration process is computationally intensive and not environmentally friendly. The goal of our work is to improve the efficiency of prompting text-to-image models, benefiting both users and service providers.

We recognize that large language models (LLMs) such as GPT-$k$ may have the possibility of generating biased content, as they may prefer the content they have seen during training, resulting in a smaller chance of showing other possibly related content. In light of this, we utilize GPT-$k$ models solely for assisting humans in the prompt editing process, and will continue to keep humans in the loop for the content creation process.

## Acknowledgments

The UCSB authors were sponsored by the Robert N. Noyce Trust. We thank the Robert N. Noyce Trust for their generous gift to the University of California via the Noyce initiative. The views and conclusions contained in this document are those of the authors and should not be interpreted as representing the sponsor.

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

## A Appendix

### A.1 DiffusionDB

DiffusionDB-2M (Wang et al., 2022) is on MIT License. It contains 2M user prompts with an average #token of $30.7 \pm 21.2$. Figure 4 shows the distribution of #token per prompt.

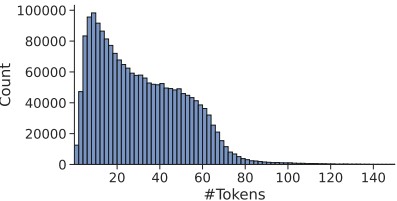

Figure 4: The distribution of the number of tokens per prompt in DiffusionDB (Wang et al., 2022).

### A.2 Hyperparameters

For the DBSCAN clustering, we set eps= $0.5$, min_samples= $1$ (note that we will filter out the clusters with only 1 sample, since it does not form a trace of edits), and metric='cosine'.

For the GPT2 model, our implementation is based on GPT2LMHeadModel on HuggingFace. During inference, we set do_sample=True, num_beams=5, max_new_tokens=80, and early_stopping=True.

For the GPT3 models, we set `temperature=0.7` and `max_tokens=256`. The other parameters just follows the default setting the OpenAI API.

For the Stable Diffusion model, our implementation is based on `CompVis/stable-diffusion-v1-4` on Huggingface. We set the random seed to $41, 42, 43$ for the three repeated runs. The `heigth`, `width`, `guidance_scale` and `num_inference_step` are kept the same as the original user specification collected in DiffusionDB for each trace of edits.

## A.3 Showcases

Table 5 presents an excerpt from an editing trace and provides examples of the four types of common edits, including `insert`, `delete`, `swap`, and `replace`. Table 6 illustrates a user editing trace where the edit of `replace` occurs in every step, and the primary subject matter of the prompt is constantly changing. Table 9 displays two sets of prompts that have been modified by the eight covered GPT-$k$ models. Figures 6 and 7 provide examples of the edits suggested by GPT-$k$ and those made by humans for head-to-head comparisons in terms of effectiveness and likelihood to be adopted by humans.

| User Input Prompt | Generated Image |
|---|---|
| `circular ornated ceiling highly detailed` | 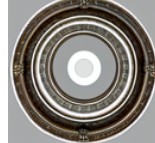 |
| `photo of an ornated circular ceiling, full of paintings of angels, centered symmetrical, highly detailed`

**SWAP**: "circular ornated" -> "ornated circular"
**INSERTION**: "full of painting of angels", "centered symmetrical" | 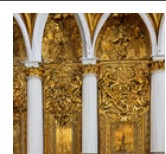 |
| `ornate marble and gold wall, full of paintings of angels, highly detailed`

**REPLACE**: "ceiling" -> "marble and gold wall"
**DELETION**: removed "centered symmetrical" | |

Table 5: Common types of edits.

## A.4 Human Evaluation

In our study, we recruited Amazon Mechanical Turk annotators to conduct human evaluations. To ensure the quality of the evaluations, we established two additional qualifications for the annotators, which included a HIT Approval Rate of higher

| User Input Prompt | Generated Image |
|---|---|
| `side profile centered painted portrait, `**`spidergwen`**`, matte painting concept art, art nouveau, beautifully backlit, swirly vibrant color lines, fantastically gaudy, aesthetic octane render, 8 k hd resolution, by ilya kuvshinov and cushart krentz and gilleard james` | 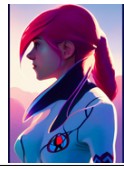 |
| `side profile centered painted portrait, `**`megan fox`**` as spidergwen, matte painting concept art, art nouveau, beautifully backlit, swirly vibrant color lines, fantastically gaudy, aesthetic octane render, 8 k hd resolution, by ilya kuvshinov and cushart krentz and gilleard james` | 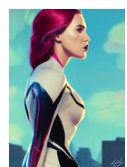 |
| `side profile centered painted portrait, `**`spiderman`**`, matte painting concept art, art nouveau, beautifully backlit, swirly vibrant color lines, fantastically gaudy, aesthetic octane render, 8 k hd resolution, by ilya kuvshinov and cushart krentz and gilleard james` | 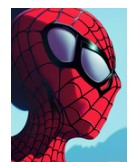 |
| `side profile centered painted portrait, `**`boba fett`**`, matte painting concept art, art nouveau, beautifully backlit, swirly vibrant color lines, fantastically gaudy, aesthetic octane render, 8 k hd resolution, by ilya kuvshinov and cushart krentz and gilleard james` | 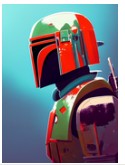 |
| `side profile centered painted portrait, `**`darth maul`**`, matte painting concept art, art nouveau, beautifully backlit, swirly vibrant color lines, fantastically gaudy, aesthetic octane render, 8 k hd resolution, by ilya kuvshinov and cushart krentz and gilleard james` | 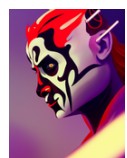 |
| `side profile centered painted portrait, `**`the punisher`**`, matte painting concept art, art nouveau, beautifully backlit, swirly vibrant color lines, fantastically gaudy, aesthetic octane render, 8 k hd resolution, by ilya kuvshinov and cushart krentz and gilleard james` | 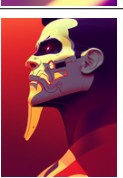 |
| `side profile centered painted portrait, `**`wolverine`**`, matte painting concept art, art nouveau, beautifully backlit, swirly vibrant color lines, fantastically gaudy, aesthetic octane render, 8 k hd resolution, by ilya kuvshinov and cushart krentz and gilleard james` | 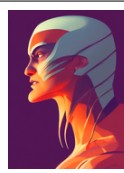 |

Table 6: A user trace where the edit of `replace` constantly occurs. The primary subject matters of the prompts are in bold, and we can see that the main subject changes from step to step in this editing trace.

than 99% and the completion of more than 100 approved HITs. The instructions provided to the MTurk annotators are displayed in Figures 6 and 7. Each HIT assignment was compensated with a payment of $0.30, and the average completion time for each assignment was approximately 2 minutes. This equates to an average hourly pay of $9-10.

## A.5 Other Image Similarity Metrics

In line with previous studies on image editing/generation (Dong et al., 2014; Han et al., 2018; Li et al., 2018; Wang et al., 2019; Zeng et al., 2021), we also employed the following metrics to evaluate the similarity between images: (1) SSIM (Wang

| Model | SSIM | | | | | PSNR | | | | |
|---|---|---|---|---|---|---|---|---|---|---|
| | $i_1$-$i_n$ | $i_{n-1}$-$i_n$ | $i'$-$i_n$ | $i'$-$i_{MS}$ | RNE | $i_1$-$i_n$ | $i_{n-1}$-$i_n$ | $i'$-$i_n$ | $i'$-$i_{MS}$ | RNE |
| gpt2-base | | | 17.60 | 22.99 | 52.32 | | | 9.35 | 10.44 | 52.92 |
| gpt2-medium | | | 17.54 | 22.96 | 52.60 | | | 9.36 | 10.45 | 53.30 |
| gpt2-large | | | 17.52 | 22.94 | 52.80 | | | 9.36 | 10.45 | 52.57 |
| gpt2-xl | 18.93 | 20.39 | 17.51 | 22.86 | 52.98 | 9.16 | 9.37 | 9.35 | 10.44 | 53.29 |
| gpt3-ada | | | 17.11 | 22.28 | 53.71 | | | 9.33 | 10.37 | 52.67 |
| gpt3-babbage | | | 16.97 | 22.12 | 52.85 | | | 9.26 | 10.30 | 52.40 |
| gpt3-curie | | | 17.41 | 22.80 | 53.51 | | | 9.29 | 10.36 | 52.25 |
| gpt3.5-davinci | | | 16.91 | 22.00 | 51.92 | | | 9.22 | 10.27 | 52.32 |

Table 7: The SSIM and PSNR scores to evaluate image similarity. Here, $i_1$, $i_{n-1}$, $i_n$ denotes the first, last-but-one, and last image in the trace of edits; $i'$ is the image generated from the modified prompt, and $i_{MS}$ is the image that is most similar to $i'$ with regard to current similarity metric.

| Mode | Prompt |
|---|---|
| In-Context $k$-Shot | The user wants to draw a picture and is deciding upon what key elements should be included. You may add, swap, or remove the modifiers, or even replace the main character. Please refer to the examples of edits and predict the elements in the final painting.

Input: {first prompt in the $k$-most related trace}
Output: {last prompt in the $k$-most related trace}

Input: {first prompt in the ($k$-1)-most related trace}
Output: {last prompt in the ($k$-1)-most related trace}

...

Input: {first prompt in the most related trace}
Output: {last prompt in the most related trace}

Input: {first prompt in current trace}
Output: |

Table 8: The prompts for GPT-3 and GPT-3.5 models.

et al., 2004) which assesses pixel-wise errors from the perspective of luminance, contrast, and structure; (2) PSNR, which compares pixels using the mean squared error.

Table 7 presents the SSIM and PSNR scores used to evaluate image similarity. For each metric, we established the similarity score between the first and last images ($i_1$-$i_n$) and the last-but-one and last images ($i_{n-1}$-$i_n$) as baselines. The similarity between the images generated from the modified prompts and the target image of the current trace ($i'$-$i_n$) is comparable to the baselines as per PSNR, however, it is lower as per SSIM. However, we also observed that the similarity between $i'$-$i_{MS}$ is significantly higher than the baselines, as per both listed metrics. This suggests that even though the image generated from the modified prompt may not be directly similar to the final target, it may be related to the intermediate steps in the editing trace. The RNE results, as per SSIM and PSNR, indicate that $i'$ is most similar to images in the middle of the trace in terms of pixel-wise comparison.

Figure 5 plots the effect of the four common types of edits. Similar to what we found in Sec-

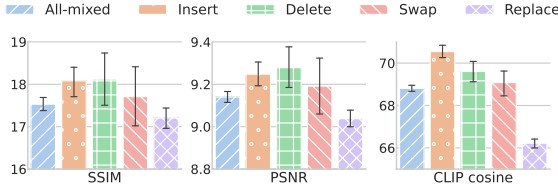

Figure 5: The SSIM, PSNR and CLIP cosine similarity scores between the image generated from the GPT-3.5-modified prompt and the last image. Results are reported on $\mathcal{S}_{eval}$ (All-mixed), $\mathcal{S}_{insert}$, $\mathcal{S}_{delete}$, $\mathcal{S}_{swap}$ and $\mathcal{S}_{replace}$.

tion 5, SSIM and PSNR have similar performances on the ablated evaluation traces – metric scores on traces that only involve insert, delete and swap are higher than or comparable to corresponding baselines. Meanwhile, we notice that replace leads to lower image similarities, which applies to all three image similarity metrics.

| | | |
|---|---|---|
| **Trace#1** | | |
| Initial Prompt in User Editing Trace | | guns and roses on a crimson and black background, intricate illustration |
| Last Prompt in User Editing Trace | | an epic guns and roses mural on a crimson and black background, intricate illustration, highly ornate, exquisite detail, rtx, 4k |
| Modified Prompt | gpt2-base | guns and roses on a crimson and black background, intricate illustration, highly detailed, simple, no jagged lines, smooth, artstation, centered artwork by obey |
| | gpt2-medium | guns and roses on a crimson and black background, intricate illustration, concept art, pixiv fanbox, art by wlop and greg rutkowski and makoto shinkai and studio ghibli and kyoto animation |
| | gpt2-large | guns and roses on a crimson and black background, intricate illustration, detailed, award - winning, trending on artstation, 4 k, 8 k |
| | gpt2-xl | guns and roses on a crimson and black background, intricate illustration |
| | gpt3-ada | funky 8 0s eurobeat, album cover, realistic, hdr, clear image |
| | gpt3-babbage | guns and roses on a crimson and black background, art by david deutsch, inspired by the movie 'Guns and Roses' |
| | gpt3-curie | guns and roses on a crimson and black background, intricate illustration, david hellyer |
| | gpt3.5-davinci | guns and roses on a crimson and black background, intricate illustration, vivid colors, concept art, artstation |
| **Trace#2** | | |
| Initial Prompt in User Editing Trace | | a bottle filled with dreams |
| Last Prompt in User Editing Trace | | a hamburger in the bottle |
| Modified Prompt | gpt2-base | a bottle filled with dreams, highly detailed, digital painting, artstation, concept art, sharp focus, illustration, art by artgerm and greg rutkowski and alphonse mucha |
| | gpt2-medium | a bottle filled with dreams, highly detailed, digital painting, artstation, concept art, sharp focus, illustration, art by artgerm and greg rutkowski and alphonse mucha |
| | gpt2-large | a bottle filled with dreams |
| | gpt2-xl | a bottle filled with dreams, art by James Jean and Wayne Barlowe, high detail, cinematic, cgsociety 8k |
| | gpt3-ada | A dream filled bottle. |
| | gpt3-babbage | a transparent plastic bottle filled with dreams, close up, 1 5 mm lens photograph |
| | gpt3-curie | a bottle filled with dreams, dreamlike background |
| | gpt3.5-davinci | a bottle of colorful dreams, vibrant colors, dreamlike elements, surrealism, magical realism, abstract painting. |

Table 9: Two sets of examples that compares the modified prompt generated by each GPT-$k$ model.

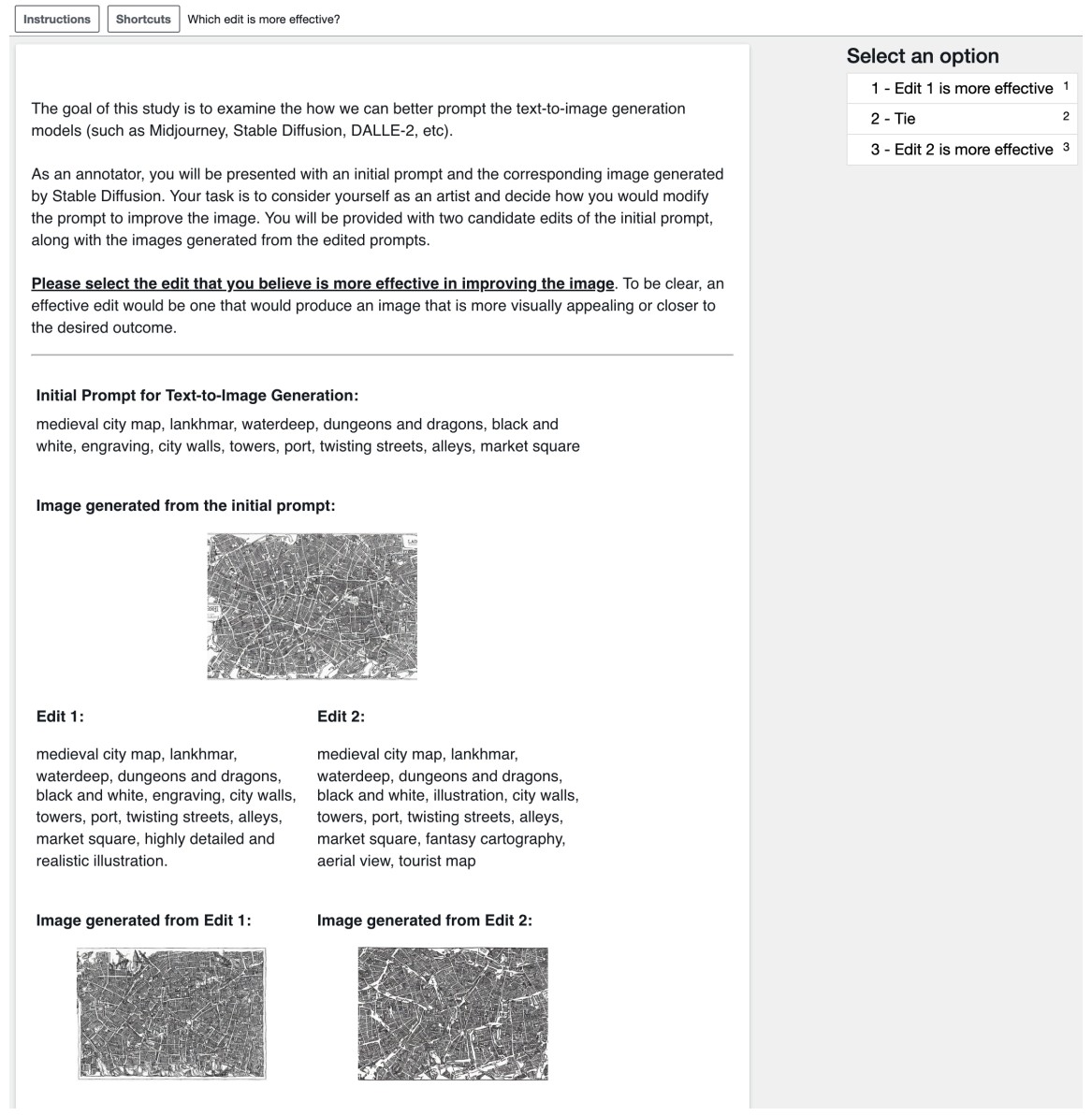

Figure 6: This screenshot illustrates the interface used in the MTurk study for a head-to-head comparison of the effectiveness of the edits suggested by the GPT-$k$ model and those made by humans. In this specific example, "Edit 1" was suggested by GPT-2-xl, while "Edit 2" represents the most similar human edit in the original editing trace.

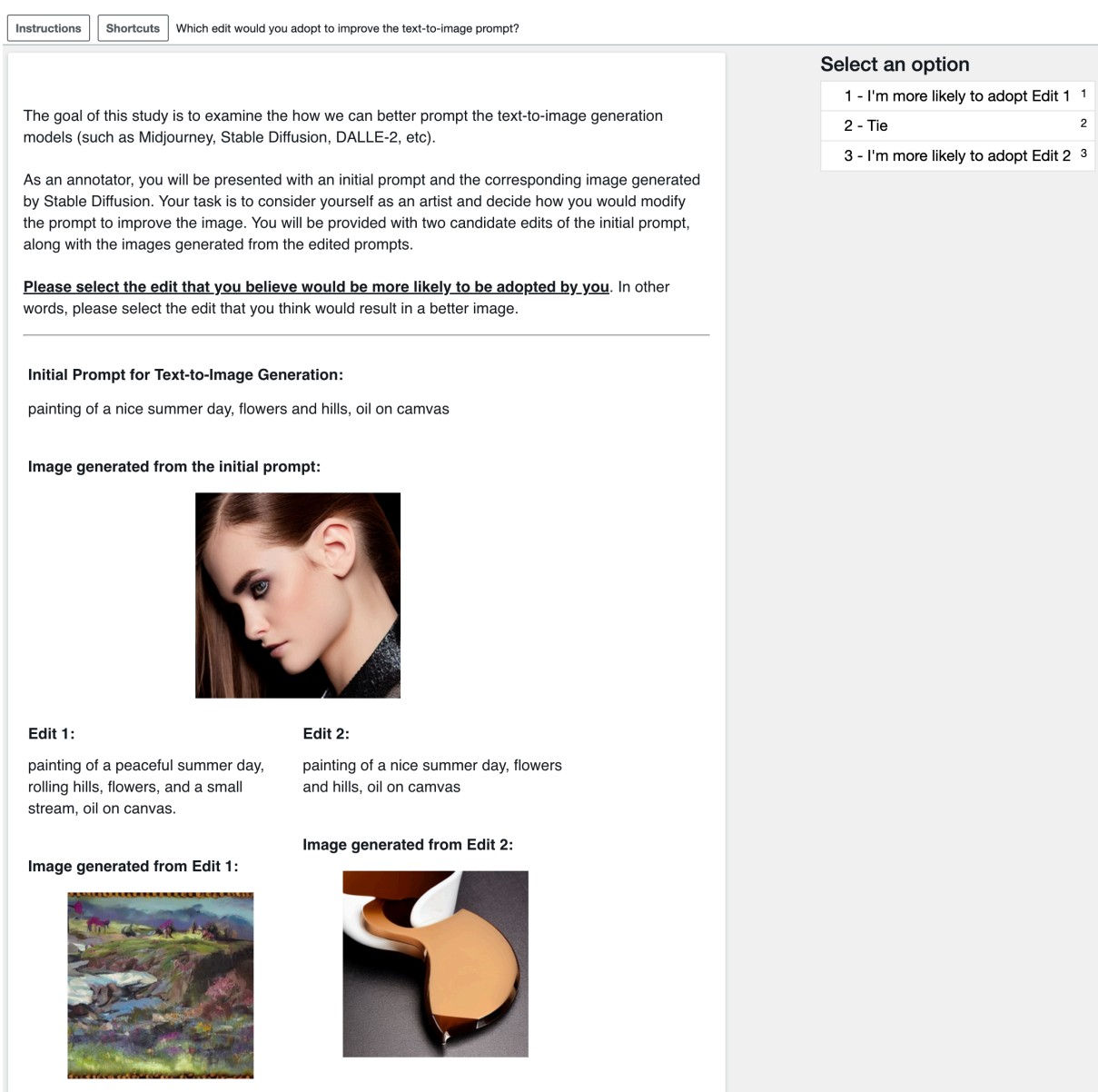

Figure 7: This screenshot illustrates the interface used in the MTurk study for a head-to-head comparison of the edits suggested by the GPT-$k$ model and those made by humans. The purpose of this evaluation was to assess the likelihood of an edit being adopted by humans. In this specific example, "Edit 1" was suggested by GPT-3.5-`davinci`, while "Edit 2" represents the most similar human edit in the original editing trace. Notice that in this example, the user actually made a typo (misspelled "canvas" into "camvas" in both the initial prompt and the prompt in Edit 2), which confused the Stable Diffusion model and thus led to unrelated renderings.

