# OpenReview forum: "Collaborative Generative AI: Integrating GPT-k for Efficient Editing in Text-to-Image Generation"
_EMNLP/2023/Conference — EMNLP 2023 Main_

### Official Review · Reviewer_zSe4 · 2023-08-04

**Soundness:** 4

**Excitement:**

4: Strong: This paper deepens the understanding of some phenomenon or lowers the barriers to an existing research direction.

**Paper Topic And Main Contributions:**

In this paper, GPT models are used to modify prompts for text-to-image generation. The authors show that, in the generation process, GPT modified prompts can reduce the number of edits by around 20%. A human study is conducted to compare the type of GPT generated edits and human edits, as well as the practicality of the usefulness of the generated prompts.

**Questions For The Authors:**

In Figure 7 in the appendix, it was quite surprising to see that Edit2 and the initial prompt are the same but ended up with both failure generations, though totally different, have you analyzed factors of such failure cases/prompts/edits? Or if any filtering has been done from the traces before performing in-context learning/prompting?

**Reasons To Accept:**

The paper is generally well written with clear motivation, solid experiments, and presented with concrete examples. It is interesting to see such analysis and how GPT models can contribute to the prompting process of text-to-image generation.

**Reasons To Reject:**

Some of the qualitative examples are a bit confusing without detailed analysis, please see the questions.

**Reproducibility:**

4: Could mostly reproduce the results, but there may be some variation because of sample variance or minor variations in their interpretation of the protocol or method.

**Reviewer Confidence:**

4: Quite sure. I tried to check the important points carefully. It's unlikely, though conceivable, that I missed something that should affect my ratings.

---

> ### Author Rebuttal · Authors · 2023-08-28
>
> Thank you for the thought-provoking feedback! Here are our responses to your questions:
>
> > **Analyze the failure case in Figure 7 in the appendix**
>
> Thanks for raising the question! For this specific example in Figure 7, the initial prompt and the prompt in Edit2 are all user-written prompts from the DiffusionDB dataset. Notice that in this example, the user actually made a typo (misspelled “canvas” into “camvas”), which confused the Stable Diffusion model and thus led to unrelated renderings. We will note this down in the revision.
>
>
> > **Is there any filtering on the traces before performing in-context learning/prompting?**
>
> To provide clarity, we want to emphasize that we refrained from implementing any filters on the clustered edit traces. It was a deliberate choice not to address potential errors (e.g., typos,  misspelled artist names) within the user prompts. Our rationale behind this decision is rooted in the fact that both the introduction of such errors and the potential subsequent rectification process are integral components of the broader "prompt editing" procedure, a focal point within the scope of our study.
>
> Illustratively, let's consider Figure 7. When the user mistakenly typed "camvas" instead of "canvas," leading to an unsatisfactory and unrelated image rendering, a pivotal question arises: how might the user modify the prompt to enhance Stable Diffusion's rendering outcome? Would they incorporate additional modifiers or alter the keywords? From our vantage point, we possess insight into the root cause behind the failure in Stable Diffusion's rendering, which is the typo within the prompt itself. Given the presence of this typo, any subsequent edits would likely still result in unsuccessful and unrelated image generation. Consequently, a more efficient and effective approach to editing involves addressing the typo initially. Notably, this is precisely the strategy employed by GPT-3.5-davinci in Edit1, underscoring how leveraging LLMs effectively enhances the efficiency of prompt editing.

---

### Official Review · Reviewer_bKnd · 2023-08-04

**Typos Grammar Style And Presentation Improvements:** 023-024 "predicting spontaneous chang…
**Soundness:** 4

**Excitement:**

4: Strong: This paper deepens the understanding of some phenomenon or lowers the barriers to an existing research direction.

**Missing References:**

Tuhin Chakrabarty, Arkadiy Saakyan, Olivia Winn, Artemis Panagopoulou, Yue Yang, Marianna Apidianaki, and Smaranda Muresan. 2023. I Spy a Metaphor: Large Language Models and Diffusion Models Co-Create Visual Metaphors. In Findings of the Association for Computational Linguistics: ACL 2023, pages 7370–7388, Toronto, Canada. Association for Computational Linguistics.

**Paper Topic And Main Contributions:**

The paper investigates the benefits of incorporating GPT-k into the text2image pipeline by using GPT-k to edit the initial t2i prompt. They then investigate how these edits perform compared to human edits. They also investigate what common edit types are made by humans compared with GPT-k and how edit types and # of edits affect the results.

**Questions For The Authors:**

A: What are hyperparameters for clustering prompts into traces of edits and how were they chosen? I was surprised to see that you have clusters with such a huge number of edits, did you inspect them to see that the clusters make sense? How many clusters in absolute numbers do you have after selecting only those that have at most 20 edits?

B: 131-135: How is the ordering for the traces determined?

C: 150-152 this does not make sense to me. Before you mentioned you only focused on 95% of the traces. How come now it is the full 100K?

D: 225-227 How exactly did you identify edit type using sequenceMatcher?

E: 233: gpt-3 are also autoregressive but you say their distribution is more similar. so the claim "GPT-2 models, due to their autoregressive training nature, have a tendency towards continual generation, resulting in a majority of edits being insert" does not hold.

**Reasons To Accept:**

1. The paper provides interesting and detailed experiments on the benefits and drawbacks of incorporating GPT-k models into the text-to-image pipeline. Some recent work was focusing on incorporating LLMs and diffusion models (see missing references) but this study provides more formal empirical investigation of this phenomenon.

2. The experiments are well-defined and executed. While additional models and settings are possible, we can already draw some preliminary conclusions from this study about the usefulness of GPT models to improve t2i generation.

**Reasons To Reject:**

1. Authors create a dataset of t2i prompt edits by clustering a dataset of t2i prompts into 100K clusters using sentence embeddings. However, no evaluation or verification of edit clusters is provided. Thus, it is hard to judge if the clusters make sense or not or if they even refer to the same image. Taking the clusters with more than just a few instructions (authors select clusters of up to 20 instructions) seems like an unrealistic setting to me and potentially increasing the chance that some of the clusters will have unrelated prompts. It would be interesting to see how smaller clusters of e.g. under 5 instructions would affect the results.

2. Each edit cluster has 1 to n edits. I find problematic how only the n-th edit of the trace was used for fine-tuning and few-shot prompts. First, as there was no guarantee that the edits were about the same image, but also the first and last edits will be very different and skipping multiple steps in the edit trace. This potentially teaches the model to hallucinate edits as it has no way to understand the intermediary steps. For example, steps in the middle might have have had different edit types completely altering the final prompt. There needs to be control for semantic/lexical similarity of first and last prompt. The experiment could also include models fine-tuned on a single edit step for a single or multiple edit types. Another way would be to vary n-th edit e.g. from 1 k=n to see the impact on i'-i_MS.

3. The human evaluation is not very clear to me. First, gpt-3-curie was chosen even though babbage has better CLIP scores, RNE (Table 2), and closer distribution to human edits (Table 4). Second, the framing of the task as choosing between prompts is very vague. For example in Figure 6 it is very hard to choose what edit would be best. It would be more clear to frame it explicitly as an image selection task (which I suspect crowdworkers did anyway).

4. In the effects of edit types experiment, the CLIP score reported is only between generated image and the last image in the edit cluster. Since each edit cluster can contain multiple types of edits, it would be more appropriate to compare to the first image on which the edit type was applied.

**Reproducibility:**

4: Could mostly reproduce the results, but there may be some variation because of sample variance or minor variations in their interpretation of the protocol or method.

**Reviewer Confidence:**

4: Quite sure. I tried to check the important points carefully. It's unlikely, though conceivable, that I missed something that should affect my ratings.

---

> ### Author Rebuttal · Authors · 2023-08-29
>
> Thank you for the careful reading and constructive comments! We will incorporate them in the revision. Here are our responses to your questions:
>
> > **Evaluation or verification of edit clusters**
>
> We manually checked 177 editing traces written by 50 users. Common types of clustering errors include:
>
> * "over-segmentation" where DBSCAN splits an editing trace into several sub-traces (12/177);
>
> * "under-segmentation" where one predicted editing trace covers several unrelated traces (4/177).
>
> Overall, more than 90% of the checked clustering results are valid and correct. Among the two types of clustering errors, we would like to note that "over-segmentation" is less problematic, since the sub-traces still center around the same topic. Consequently, the feasibility of deriving meaningful edits endures in this scenario.
>
> We’ll incorporate such results in the revision.
>
> > **Why only using the n-th edit of the trace for fine-tuning and few-shot prompts?**
>
> Thanks for raising the question on the experimental design. We are more than delighted to provide a deeper insight into our rationale.
>
> In the submission, we directly employ the last prompt in the editing trace for tuning and in-context learning. This decision emanates from our primary objective of conducting more *efficient* prompt editing. In line with this goal, our intention is to expedite the evolution of the prompt toward its final form, seeking to minimize the number of necessary edits.
>
> It is indeed acknowledged that this approach might bypass several intermediate editing steps. However, we wish to emphasize that this particular choice is entirely intentional. By employing this strategy, we are intentionally encouraging the language model to proficiently skip or amalgamate redundant editing stages, in turn striving for an elevated efficiency in the prompt editing process.
>
>
> > **How did you select models for human evaluation?**
>
> Thank you for inquiring about our model selection process for the human evaluation phase. Specifically, we made a deliberate decision to select one model from each of the model categories encompassing GPT-2, GPT-3, and GPT-3.5, as outlined in Table 1. Within each distinct model category, our focus gravitated towards the selection of the most extensive model available. Thus, we choose the largest model with the most parameters (`gpt2-xl` for GPT-2, `text-curie-001` for GPT-3, and `text-davinci-003` for GPT-3.5).
>
> > **Framing of the human evaluation**
>
> We invite the annotators to conduct head-to-head comparisons, and provide three options of “win/lose/tie”. If the annotator finds it hard to decide which edit is better (e.g., Figure 6), then the annotator could choose “tie”.
>
>
>
> > **Compare the modified prompt to the first image on which the edit type was applied**
>
> In the submission, we compare the modified prompt to the last prompt in the editing trace. This is because we attempt to conduct more efficient prompt editing, which means we want the modified prompt to be as similar to the final prompt as possible.
>
> We are happy to provide more results comparing the modified prompts of each edit type to the original prompt. In the following table, we list out the CLIP similarity between the modified prompt $i’$ and the first prompt in the trace of edits $i_1$, as an extension to the results in Figure 2.
>
> | Edit Type | $i’$ vs. $i_1$ | $i’$ vs. $i_n$ (results in Figure 2) |
> |-----------|----------------|--------------------------------------|
> | All-mixed | 75.36          | 68.81                                |
> | Insert    | 76.08          | 70.54                                |
> | Delete    | 76.08          | 69.61                                |
> | Swap      | 75.98          | 69.08                                |
> | Replace   | 74.78          | 66.22                                |
>
> We can see that the edit types of `insert`, `delete` and `swap` have comparable impacts on raising the CLIP image similarity, while the edit of `replace` lowers the CLIP image similarity. The trend aligns with the results comparing the modified prompt with the last prompt as discussed in the submission (Figure 2, line 243-270).
>
> > **Question A: Hyperparameters for clustering prompts into trace of edits**
>
> For the DBSCAN clustering, we set:
>
> *  `eps = 0.5`. The `eps` is the maximum distance between two samples for one to be considered as in the neighborhood of the other. The value of eps of 0.5 is the default value as suggested by the package readme. After varying the eps value and empirically checking the clustering results, we decided to continue with the default eps value of 0.5.
>
> * `min_samples=1`. The `min_samples` is the number of samples (or total weight) in a neighborhood for a point to be considered as a core point. Note that we will filter out the clusters with only 1 sample, since it does not form a trace of edits.
>
> * `metric ='cosine'`, which is the metric to use when calculating the distance between instances in a feature array.
>
>
>
> > **Question A & C: How many clusters in absolute numbers do you have after selecting only those that have at most 20 edits?**
>
> Thanks for asking! We received 100433 traces of edits after clustering. All of the 100k traces were used in our study. The validation set (30k) was randomly sampled from the traces with less than 20 edits.
>
> What we meant by *“in the following experiments, we consider traces with at most 20 edits”* (line 123-124) is that the evaluation results were reported on traces with at most 20 edits.
>
> We sincerely apologize for causing the confusion, and would definitely clarify this in the revision.
>
>
> > **Question B: How is the ordering for the traces determined?**
>
> The DiffusionDB dataset collects the timestamp of each user prompt. We order the prompts by corresponding timestamps in each trace of edits. We will clarify this in the revision.
>
>
> > **Question D: How exactly did you identify edit type using SequenceMatcher?**
>
> Thanks for asking! We will clean and release the code used in this study upon acceptance. Since we are not allowed to put external links and considering the fact that there is no word limit in the rebuttal, we decided to directly paste the script we used here:
>
>
> ```python
> import string
> from difflib import SequenceMatcher
> from nltk.tokenize import word_tokenize
> from nltk.corpus import stopwords
> from collections import defaultdict
>
> stopword_list = set(stopwords.words('english'))
>
> def _preprocess(sent):
>    sent = sent.lower().translate(str.maketrans('', '', string.punctuation))
>    word_tokens = word_tokenize(sent)
>    filtered_tokens = [w for w in word_tokens if not w in stopword_list]
>    return filtered_tokens
>
>
> def find_edit_type(a, b):
>    edit_type = defaultdict(lambda:0)
>    a = _preprocess(a)
>    b = _preprocess(b)
>    s = SequenceMatcher(None, a, b)
>    opcode_list = list(s.get_opcodes())
>
>    # check if there is any equal sub-sequence
>    # if there is no equal, then not valid
>    no_equal = True
>    for tag, _, _, _, _ in opcode_list:
>        if tag == 'equal':
>            no_equal = False
>            break
>    if no_equal:  # not valid, early return with all zeros
>        return edit_type
>
>    # check the last op
>    tag, i1, i2, j1, j2 = opcode_list[-1]
>    if tag == 'replace':
>        if i1 == i2 and j1 < j2:
>            edit_type['insert'] += 1
>        elif i1 < i2 and j1 == j2:
>            edit_type['delete'] += 1
>        else:
>            edit_type['replace'] += 1
>    else:
>        edit_type[tag] += 1
>
>    # check the previous ops
>    a_str_list = []
>    b_str_list = []
>    for idx, (tag, i1, i2, j1, j2) in enumerate(opcode_list):
>        if idx < len(opcode_list)-1:
>            edit_type[tag] += 1
>
>        # check if there is any swapping
>        if tag != 'equal':
>            a_str = ' '.join(a[i1:i2])
>            b_str = ' '.join(b[j1:j2])
>            if a_str != '':
>                for _str in b_str_list:
>                    if _str.find(a_str) > -1:
>                        edit_type['swap'] += 1
>                        edit_type[tag] -= 1
>                        break
>            if b_str != '':
>                for _str in a_str_list:
>                    if _str.find(b_str) > -1:
>                        edit_type['swap'] += 1
>                        edit_type[tag] -= 1
>                        break
>
>            if a_str != '':
>                a_str_list.append(a_str)
>            if b_str != '':
>                b_str_list.append(b_str)
>
>    return edit_type
>
>
> def compare_prompts_and_find_edits(prompt_list):
>    all_edit_types = defaultdict(lambda:0)
>
>    for a, b in zip(prompt_list[:-1], prompt_list[1:]):
>        edit_count = find_edit_type(a, b)
>        for edit_type, count in edit_count.items():
>            all_edit_types[edit_type] += count
>
>    for edit_type in ['insert', 'swap', 'delete', 'replace']:
>        print(f'{edit_type}: {all_edit_types[edit_type]}')
>
>    return all_edit_types
>
>
> if __name__ == '__main__':
>    edit_trace = [
>        'The quick brown fox jumps over the lazy dog',
>        'Look! The brown quick fox jumps over the dog!',
>    ]
>    compare_prompts_and_find_edits(edit_trace)
>
>    """
>    Output:
>        insert: 1
>        swap: 1
>        delete: 1
>        replace: 0
>    """
> ```
>
> > **Question E: Why GPT-2 and GPT-3 have different patterns of edits?**
>
> Thanks for pointing out the loose statement! We will provide a more rigorous analysis in the revision.
>
> We agree that both GPT-2 and GPT-3 models undergo training in an autoregressive fashion. However, a significant difference between the two types of models is that GPT-3 demonstrates emergent ability, which enables it to engage in in-context learning. Unlike GPT-2, which solely receives the specific prompt requiring editing during testing, GPT-3 is also provided with additional editing instances as supportive exemplars for in-context learning. This emergent ability within GPT-3 is a pivotal factor that diverges from mere plain continuation. Consequently, while GPT-2 indeed adheres to its autoregressive inference manner, the emergent capability of GPT-3 enables it to attempt other edit types that simulate human-like editing behavior, as illustrated by the in-context examples.
>
>
> > **Missing reference**
>
> Thanks for pointing us to this publication on related topics! We will definitely include this in the revision.
>
> > **Typos**
>
> Thanks for pointing out! We apologize for the oversight and will fix the writing issues in the revision.

---

### Official Review · Reviewer_AHM3 · 2023-08-05

**Soundness:** 4

**Excitement:**

4: Strong: This paper deepens the understanding of some phenomenon or lowers the barriers to an existing research direction.

**Paper Topic And Main Contributions:**

The paper investigates the concept of "prompt editing" associated with text-to-image generation, in which the users apply successive edits to their prompts in order to generate their desired image. It does so by analyzing a dataset of user prompts scraped from the official Stable Diffusion Discord server and employing various GPT-based models to improve the prompt editing process. In their experiments the authors observe that while the GPT-based models seem to mostly insert modifiers, human users tend replace words and phrases, yielding a significantly different image in the end. Furthermore, it also seems that editing the prompt via GPT-based models results in images more similar to intermediate edits made by the users rather than the final image, suggesting that the GPT-based models may be more effectively used to reduce the number of edits necessary to arrive at the final image.

**Questions For The Authors:**

**Question A**: Would you mind providing some more details on how you used the GPT-3 models in the evaluation, especially what parameters were provided to the API and also how expensive was it to execute the experiments described in the paper?

**Question B**: Is there any specific reason why the `gpt-3.5-turbo` model wasn't used in the experiments as opposed to `text-davinci-003`, which is both more expensive and which the OpenAI's docs now consider legacy?

**Question C**:  Would you be able to provide any intuition as to what extent are the GPT-3 results robust to changes of the prompt shown in Table 8?

**Question D**: Similarly to the previous question, how did you arrive at choosing 16 shots for the in-context learning and to what extent would varying this number change the results?

**Reasons To Accept:**

- An analysis of the prompt edits made by users at the Stable Diffusion Discord server
- Exploration of the difference between the images generated by user-edited vs. GPT-generated prompts
- Evaluation of various GPT-based models on this task, with the GPT-2 model family being finetuned and GPT-3 model family being prompted
- Human evaluation of the prompts generated/edited by the GPT-based models, showing them to be better or comparable to user edited prompts
- An ablation study on the edit types (insert/delete/swap/replace) on the generated output image

**Reasons To Reject:**

The paper is well written and hence there are only a few reasons why this paper might be rejected.

Perhaps the strongest one is the fact that while the GPT-based models are a key part of the analysis, except for the prompt the authors used the paper doesn't contain any further details on how it was used as a metric. Furthermore, the OpenAI's APIs provide various parameters such as `temperature`, `top_p`, `presence_penalty` and `frequency_penalty`, which are not specified in the paper, further complicating potential future replication efforts. This should also be applicable to GPT-2 models that were publicly released and are not available only via an API (e.g. the HuggingFace Transformers provide APIs for specifying similar parameters).

A similar issue can be found with the Stable Diffusion model that was used for image generation. It is not obvious what parameters, except for the prompts, were varied when the images were generated, and to what extent would changing these have an impact on the presented results.

There were also two relatively minor inconsistencies we found in the paper which unfortunately detract from the quality of the paper.

The first can be found on line 151 when they authors mention "We split the 100k trace of edits into two halves," but the next part of the sentence says "with 30k traces used for evaluations and the remaining 70k serving as holdout set". When splitting something into two halves, it would be reasonable to expect a 50:50 split but that is not what we observe here. It is therefore unclear which of the potential interpretations the authors meant in this sentence.

The second can be found at the discussion of edit types which starts on line 217. When using SequenceMatcher, it is unclear how are the edits classified. If for instance one generated prompt both swaps the order of words, inserts a few new words towards the beginning and removes a few words from the end of the prompt, which category would such a change fall into? Would it be all of them or just one of them or neither?

**Reproducibility:**

3: Could reproduce the results with some difficulty. The settings of parameters are underspecified or subjectively determined; the training/evaluation data are not widely available.

**Reviewer Confidence:**

4: Quite sure. I tried to check the important points carefully. It's unlikely, though conceivable, that I missed something that should affect my ratings.

**Typos Grammar Style And Presentation Improvements:**

- line 051: "issue" -- it would be best to replace this word with say "phenomenon" or "figure"
- line 060: "LLMs and T2I models are trained on different modalities and architectures" -- although the models do use different architectures, it is not clear what training on architectures would refer to
- line 138: "denotes" -> "denote"

---

> ### Author Rebuttal · Authors · 2023-08-28
>
> Thank you for the careful and insightful review. We appreciate your feedback and will take them into account when revising our paper.  Here are our responses to your questions:
>
> > **Question A: Detailed parameter setup for GPT models and Stable Diffusion**
>
> For the GPT2 model, our implementation is based on `GPT2LMHeadModel` on HuggingFace. During inference, we set `do_sample=True`, `num_beams=5`, `max_new_tokens=80`, and `early_stopping=True`.
>
> For the GPT3 models, we set `temperature=0.7` and `max_tokens=256`. The other parameters just follows the default setting the OpenAI API.
>
> For the Stable Diffusion model, our implementation is based on `CompVis/stable-diffusion-v1-4` on Huggingface. We set the random seed to `41,42,43` for the three repeated runs. The `heigth`, `width`, `guidance_scale` and `num_inference_step` are kept the same as the original user specification collected in DiffusionDB for each trace of edits.
>
> > **Train/validation split**
>
> Thanks for pointing out! The train/validation split is 70k vs. 30k. We apologize for causing the confusion and will definitely fix this in the revision.
>
> > **How to classify the edit category if multiple types of edits appear?**
>
> We will count the appearance of each edit. As an illustrative example, if we compare the following two prompts:
>
> * prompt1: *'The quick brown fox jumps over the lazy dog'*
>
> * prompt2: *'Look! The brown quick fox jumps over the dog!'*
>
> Then there are three types of edits from prompt1 to prompt2:
> * insert x1: (prompt2 adds “Look”)
>
> * swap x1: (prompt1 “quick brown” -> prompt2 “brown quick”)
>
> * delete x1: (prompt2 removes “lazy”)
>
>
>
> > **Question B: Why wasn’t gpt-3.5-turbo used?**
>
> Thanks for asking! The main experiments of this submission were conducted during January 2023, by which time gpt-3.5-turbo has not been released yet (release date: March 1st, 2023).
>
>
> > **Question C: Would you be able to provide any intuition as to what extent are the GPT-3 results robust to changes of the prompt shown in Table 8?**
>
> Thanks for asking! In our pilot study, we experimented with several different framings of the instructional prompts for GPT-3, such as:
>
> * framing the task as summarization: *“Summarize the following prompt to around {num_kept_tokens} tokens.”*
>
> * framing the task as rewriting: *“Rewrite the following prompt to list out the key elements in a painting, no numbering, connect with comma.”*
>
> * describing the scenario but not specifying the type of edits: *“The user wants to draw a picture, but is not satisfied with the current drawing and keeps modifying the key elements in it. The trace of edits is provided. Please predict the elements in the final painting.”*
>
> We notice that when using the framing of summarization or rewriting, GPT-3 tends to summarize/rearrange the existing keywords that appeared in the original prompts, which is quite different from human edits.
>
> With the third framing, the GPT-3 predictions are more flexible, but we notice that the edit of “insertion” occurs more often than other types of edits. As a result, we further add a line of instruction specifying the possible types of edits: *“You may add, swap, or remove the modifiers, or even replace the main character”*, which makes up the instruction used in the submission, as listed out in Table 8.
>
>
> > **Question D: How did you arrive at choosing 16 shots for the in-context learning and to what extent would varying this number change the results?**
>
> We conduct pilot studies varying the number of prompting examples for GPT3 models, and choosing k=16 is based on empirical evidence.
>
> The following table shows prompting GPT-3.5-davinci with k=2, 4, 8, 16, 32 examples, as an extension to the results in Table 2 in the submission. We can see that the change of k does not have a significant impact on the final results.
>
> | k  | $i’$ - $i_n$ | $i’$ - $i_{MS}$ | RNE (%) |
> |----|--------------|-----------------|---------|
> | 2  | 68.31        | 77.33           | 68.97   |
> | 4  | 68.66        | 77.85           | 69.47   |
> | 8  | 68.84        | 78.07           | 69.53   |
> | 16 | 68.79        | 78.09           | 69.22   |
> | 32 | 68.90        | 78.02           | 69.17   |
>
>
> > **Typos**
>
> Thanks for pointing out! We apologize for the oversight and will fix the writing issues in the revision.

---

### Meta-Review · Area_Chair_U7bK · 2023-09-18

**Recommendation:** 4

**Metareview:**

This paper investigates the capabilities and limitations of using GPT-k to edit prompts for text-to-image generation. It focuses on understanding the capability of the GPT-k models for text-to-image generation rather than trying to improve it. Overall this is a solid paper. Reviewers found this paper well written and easy to follow. 2 reviewers found the experiments well-defined and executed with concrete examples and good details. One reviewer commented on the lacking of qualitative evaluation and verification of the edit clusters. The authors were able to provide the manual verification results and the cluster error breakdown in the response. This is an important aspect for supporting the finding presented in this paper. The authors should consider add these details in the later version.

---

### Decision · Program_Chairs · 2023-10-07

**Decision:**

Accept-Main

**Comment:**

This paper investigates the capabilities and limitations of using GPT-k to edit prompts for text-to-image generation. It focuses on understanding the capability of the GPT-k models for text-to-image generation rather than trying to improve it. Overall this is a solid paper. Reviewers found this paper well written and easy to follow. 2 reviewers found the experiments well-defined and executed with concrete examples and good details. One reviewer commented on the lacking of qualitative evaluation and verification of the edit clusters. The authors were able to provide the manual verification results and the cluster error breakdown in the response. This is an important aspect for supporting the finding presented in this paper. The authors should consider add these details in the later version.